# Phylogenetic Analysis and Genetic Structure of Schlegel’s Japanese Gecko (*Gekko japonicus*) from China Based on Mitochondrial DNA Sequences

**DOI:** 10.3390/genes14010018

**Published:** 2022-12-21

**Authors:** Longjie Xia, Fengna Cai, Shasha Chen, Yao Cai, Kaiya Zhou, Jie Yan, Peng Li

**Affiliations:** 1Jiangsu Key Laboratory for Biodiversity and Biotechnology, College of Life Sciences, Nanjing Normal University, Nanjing 210023, China; 2School of Food Science, Nanjing Xiaozhuang University, Nanjing 211171, China

**Keywords:** *Gekko japonicus*, mitochondrial DNA, COI, phylogenetic analysis, genetic diversity

## Abstract

*Gekko japonicus*, i.e., Schlegel’s Japanese Gecko, is an important species which is widely distributed in East Asia. However, the information about population genetics of this species from China remains unclear. To address this issue, we used sequences from a fragment of the mitochondrial protein-coding gene cytochrome c oxidase I to estimate genetic diversity, genetic structure, and historical demography of *G. japonicus* populations from China. Phylogenetic analysis indicated that *G. japonicus* had a close relationship with *Gekko wenxianensis*. A total of 14 haplotypes were obtained, of which haplotype 1 was the most common and widely distributed. The genetic diversity of *G. japonicus* was comparatively low across different geographic populations. The populations of *G. japonicus* were divided into four groups which exhibited low levels of genetic differentiation, and expressed an unclear pattern of population structuring. In addition, potential population expansion of *G. japonicus* has occurred as well. Overall, these results demonstrate that the populations of *G. japonicus* reveal low genetic diversity in China, which is attributed to the founder and bottleneck events among populations. Our results will provide meaningful information on the population genetics of *G. japonicus* and will provide some insights into the study of origin of populations.

## 1. Introduction

*G. japonicus*, known as Schlegel’s Japanese Gecko, is a nocturnal gecko which was reported to be distributed in China, Japan, and Korea [1]. Currently, studies on a variety of topics including population distribution, morphological characteristics, daily activity pattern, selection of oviposition site, and mitochondrial genome, have been performed on the *G. japonicus* [2]. Undoubtedly, the specialized body mechanism and ecological role make *G. japonicus* an excellent flagship species for many issues in ecological balance and biological evolution. Predictions of present and future distribution studies which were based on MaxEnt modeling, suggest that suitable habitats are currently located in coastal cities of Japan, China, and Korea, as well as in isolated patches of inland China. Due to climate change, suitable habitats are expected to shrink along coastlines, particularly at the coastal edge of climate change zones [3]. Recent concerns regarding the area of origin and genetic relationships among populations of *G. japonicus* are growing, as are the range expansions of *G. japonicus* in Korea and Japan [3]. Genetic diversity and inferred dispersal history demonstrated that the populations of *G. japonicus* from China is inferred to be the source population, which had higher genetic diversity and more private alleles compared to the populations of *G. japonicus* in Japan and Korea [4]. In brief, the genetic structure and distribution pattern of *G. japonicus* in China will provide important information about the origin and evolutionary of this species. Nevertheless, to date, information of its populations, particularly from China, is unavailable. The genetic diversity, genetic structure, and historical demography of *G. japonicus* from China remains unclear.

The current distribution of *G. japonicus* in China includes the eastern coast extending westward to eastern Sichuan Province, northward to southern Shaanxi, and Gansu Provinces, which overlaps with *Gekko scabridus*, *Gekko hokouensis*, *Gekko subpalmatus*, and *Gekko chinensis* [1,3]. *G. japonicus* and other species of gecko could be separated by morphological and molecular manners [1]. In some cases, however, species complex may display few or no phenotypic differences, making them impossible to be distinguished based on morphology [5,6]. By definition, species that are morphologically indistinguishable, but belong to distinct evolutionary lineages are referred to as cryptic species [7,8,9,10]. Molecular data, especially mitochondrial DNA (mtDNA), has shown to be an efficient tool for identification of species, owing to its relatively rapid evolution rate and lack of recombination [11,12]. Meanwhile, the mtDNA has been broadly applied in population genetics and phylogenetic studies [12,13]. Among these mitochondrial genes, the cytochrome c oxidase subunit 1 (COI) has been vastly employed in identification of species, study of genetic diversity and genetic structure on different lizard species [4,14]. This approach has aided the rapid identification of species and analysis of diversity patterns in diverse groups [15,16]. Specifically, COI genes could provide valuable information for understanding the population structure and variations.

To detect the genetic structure and diversity of *G. japonicus*, 325 specimens were collected from 37 sampling sites in China. The sequences of COI genes were obtained from these 325 specimens in China. Subsequently, phylogenetic analyses of *G. japonicus* with other species of *Gekko* were conducted to clarify the genetic relationship among the genus *Gekko*, which is necessary for the further understanding of the evolutionary history of *Gekko*. Genetic structure, genetic diversity, and historical demography of *G. japonicus* populations were determined. To our knowledge, this study is the first to understand the genetic structure, genetic diversity, and historical demography of *G. japonicus* populations in China based on the mitochondrial gene COI.

## 2. Materials and Methods

### 2.1. Sampling

A total of 325 individuals were collected from 37 sampling sites that covered most of the range of *G. japonicus* in China from 2008 to 2017 (Figure 1). They were identified as *G. japonicus* and preserved in 95% ethanol for genomic DNA extraction. Geographic coordinates of sampling sites were recorded with a Geographic Positioning System unit (Garmin GPSmap 60CSX, Shanghai, GARMIN, USA) using map datum WGS84. The sample information that was used is shown in Appendix A.

### 2.2. DNA Extraction and Sequencing

Genomic DNA was extracted from liver tissue using DNA extraction kit (Qingke, Nanjing, China) according to the manufacturer’s instructions, and then the quality and integrity of the DNA samples were checked by agarose gel electrophoresis. The extracted DNA was stored at −20 °C. A partial fragment of the COI was amplified by the polymerase chain reaction (PCR) using primers: VF1-d (5′-TTCTCAACCAACCACAARGAYATYGG-3′) and VR1-d (5′-TAGACTTCTGGGTGGCCRAARAAYCA-3′) [17]. Amplifications were carried out in 20 µL reaction volumes containing 10 µL of 2 × *Taq* PCR Master Mix Ⅱ (TIANGEN, Beijing, China), 0.7 µL of genomic DNA, 0.4 µL of each primer, and 8.5 µL of deionized water. The cycle profiles for the PCR were as follows: initial denaturation of 3 min at 95 °C, followed by 5 cycles of 30 s at 95 °C, annealing for 1 min at 45 °C, an initial extension for 2 min at 72 °C; followed by 35 cycles of 30 s at 95 °C, annealing for 1 min at 51 °C, an initial extension for 2 min at 72 °C; and final extension for 5 min at 72 °C. PCR products were visualized on a 1% agarose gel stained with 4S Red Plus, Bi-directional sequencing was performed using the primers RepCOI-F (5′-TNTTMTCAACANACCACAAAGA-3′) and RepCOI-R (5′-ACTTCTGGRTGKCCAAARAATCA-3′) [18] by Nanjing Qingke Biology Co., Ltd. (Nanjing, China).

### 2.3. Sequence Alignment and Phylogenetic Analyses

We searched GenBank (National Center for Biotechnology Information, https://www.ncbi.nlm.nih.gov (accessed on 1 February 2022)) for published COI of some species of the genus *Gekko*. Then forty-four COI sequences were retrieved and downloaded for further analyses (Appendix A). All 325 individuals were successfully amplified for COI. The chromatograms of each sequence were proofread and then assembled using SeqMan 12.3 [19]. Multiple sequences were aligned and trimmed using ClustalW implemented in MEGA 7.0 [20], aligned with the COI sequences of part species of *G. japonicus* group (*G. auriverrucosus*, *G. chinensis*, *G. hokouensis*, *G. japonicus*, *G. scabridus*, *G. subpalmatus*, *G. swinhonis*, *G. wenxianensis*), *Gekko gecko* group (*Gekko gecko*) and *Gekko monarchus* group (*Gekko kikuchii*), and *Hemidactylus dracaenacolus* and *H. granti* (Appendix A), in which *H. dracaenacolus* and *H. granti* were used as outgroups for the phylogenetic analyses of the mtDNA. 

The K2P model was used to calculate genetic distances using MEGA 7.0 [21]. For phylogenetic reconstruction, we performed maximum likelihood (ML) and Bayesian inference (BI) by using IQ TREE 1.6.12 [22] and MrBayes 3.2.6 [23]. PartitionFinder2 [24] was run to determine the appropriate model of molecular evolution in a likelihood ratio test framework based on the Akaike Information Criterion (AIC) and Bayesian Information Criterion (BIC) for the ML and BI method, respectively. For the maximum likelihood (ML), bootstrap analyses were performed on 1000 full heuristic replicates [25]. For the Bayesian inference, MarkovChain Monte Carlo chains were run for 10,000,000 generations (sampled every 1000 generations) to allow adequate time for convergence. The first 25% of the sampled trees were considered burn in. The final tree was visualized and edited by ITOL (https://itol.embl.de/ (accessed on 1 February 2022)).

### 2.4. Population Genetic Analysis

To identify similar groups of populations and to evaluate the amount of genetic variation among the partitions, a spatial analysis of molecular variance was conducted in SAMOVA v1.092 [26]. Several runs were performed using increasing numbers of groups (K = 1–20) and 100 annealing simulations for each K. In each run, populations were clustered into genetically and geographically homogenous groups. The number of groups was chosen so as to maximize genetic differentiation among the groups (F_CT_) [26]. The numbers of haplotypes, haplotype diversity (*h*) [27], and nucleotide diversity (*π*) for each population were estimated using DnaSP 6.0 program [28]. Spatial genetic variation was also quantified by estimating analog of F_ST_ using analysis of molecular variance (AMOVA) [29]. In addition, to identify the relationship among haplotypes, the mtDNA haplotypes of *G. japonicus* were estimated from a TCS network using PopART v1.7 [30]. The relationships of haplotypes were further elucidated using ML analysis implemented in IQTREE. The confidence levels at nodes after 1000 repetitions employed by the bootstrap method. The historical population demography was evaluated using Tajima’s *D* statistic [31], Fu’s *Fs* [32] and mismatch distribution test [33]. Mismatch distributions for the species were calculated with the expected frequency based on a population growth-decline model. The sum of squared deviations (SSDs) between observed and expected mismatch distribution and the raggedness index [34] were calculated to test the null hypothesis of spatial expansion using Arlequin 3.5 [35].

## 3. Results

### 3.1. Sequence Information

A total of 325 mitochondrial COI gene sequences of *G. japonicus* were obtained and further analyzed. The sequences were deposited in GenBank under project SUB12126169. The average frequencies of T, C, A, and G were 28.6, 30.7, 21.1, and 19.6%, respectively. The sequenced region contained 564 conserved sites, 75 variable sites, and 16 parsimony informative sites. The ratio of transform to transition (Ts/Tv) based on the K2P base substitution model is 2.2.

### 3.2. Phylogenetic Analysis

In the study, the ML tree and BI tree showed that all species formed a monophyletic group with high bootstrap and posterior probability values (Figure 2). Species of the *G. japonicus* group cluster well together and form sister groups with *Gekko gecko* group and *Gekko monarchus* group. ML tree and BI tree revealed that the *G. swinhonis* samples were separated into A1, A2, and B. Both in BI and ML tree, *G. japonicus* cluster formed a sister group with *G. wenxianensis*, which suggested that *G. japonicus* has a close relationship with *G. wenxianensis*. Then, *G. swinhonis* was clustered with *G. auriverrucosus* and the two formed a sister group with *G. subpalmatus*. In addition, a total of 12 individuals were collected from Anqing, Anhui Province, of which 11 individuals were clustered into *G. japonicus*, undoubtedly. AHAQ06 draws our attention by forming a sister group with all the *G. japonicus*. Subsequently, according to the COI genes of the AHAQ06 and 10 gecko species of China, interspecific and intraspecific genetic distance was calculated to further illustrate the reliability of the phylogenetic results (Table 1). Except for AHAQ06, the results indicated that the genetic distance between the *G. japonicus* and *G. wenxianensis* was the smallest (21.7–23.5), the genetic distance between the *G. subpalmatus* and *G. gecko* was the largest (42.0–44.9). Meanwhile, the genetic distance within the *G. auriverrucosus* was the smallest (0.3 ± 0.2) and the genetic distance within the *G. scabridus* was the largest (4.3 ± 0.6). Moreover, the genetic distance between AHAQ06 and *G. japonicus* had reached 12.0–12.9. The genetic distances largely coincide with the phylogenetic results. Therefore, there are large differences between AHAQ06 and *G. japonicas* at the molecular level, which indicate that AHAQ06 is a special individual to be further studied. Hence, AHAQ06 is excluded from the population-level data set. A total of 324 mitochondrial COI gene sequences was depicted in further population genetic analysis.

### 3.3. Genetic Diversity, Genetic Structure and Historical Demography of G. japonicus Population

#### 3.3.1. Genetic Diversity

Fourteen haplotypes were identified among the 324 sequences (AHAQ06 are excluded) in *G. japonicus* (Appendix A). The most frequent haplotype detected was Hap 1, shared by 233 individuals, followed by haplotypes Hap 5 shared by 55 individuals. The least frequent haplotypes were Hap 2, Hap 7, Hap 9, Hap 10, Hap 11, Hap 13, and Hap 14, each of which were recorded and designated as “private haplotypes” (Table 2). Geographic populations with a sample size of more than 5 were selected during the genetic diversity analysis to avoid errors, where possible. The population genetic analysis revealed that the nucleotide diversity (π) and average number of nucleotide differences (k) were highest in the SXYX population, based on COI genes at 0.00241, 1.600. Haplotype diversity between all sequences obtained ranged from 0 to 0.591. The highest *h* was the HNYZ population (*h* = 0.591), whereas the lowest was the HNYY and HNCB populations (*h* = 0). In general, total genetic diversity values were low between populations (*π* = 0.00176, *h* = 0.453), compared to *Chilabothrus inornatus* [36].

#### 3.3.2. Genetic Structure

According to the COI data, SAMOVA revealed that the proportion of the total genetic differences between groups (F_CT_) was the highest (0.39527) when K = 4 (Appendix A). Four groups were separated as follows: A: Yongan, Fujian, Yangshuo, Guangxi, Huaihua, Hunan; B: Yangxian, Shaanxi. C: Anhui (Anqing, Lu’an, Wuhu), Fujian Nanping, Guangzhou Guiding, Guizhou (Huaxi, Longli, Libo), Guangxi (Guilin, Longsheng), Hubei (Jingmen, Wuhan), Hunan (Chengbu, Changde, Daoxian, Huayuan, Huayuan, Shuangpai, Shaoyang, Tong, Xinhuang, Yueyang, Yongzhou, Xinning, Yongzhou Lengshuitan, Zhuzhou, Yongzhou Qingtang Yueyyan Forest Farm), Jiangsu Rugao, Jiangxi Longnan, Zhejiang (Yongzhou Qingtang Yueyyan Forest Farm). Zhoushan, Lishui). D: Yongfu, Guangxi. The total variation was F_ST_ = 0.40362 and the intragroup variation was F_SC_ = 0.01382 when K = 4. The AMOVA results revealed that the source of genetic differences emerged primarily from within the populations (Table 3). The groups differed significantly from each other (F_CT_ = 0.39527) and were responsible for 39.527% of the variance. The non-significant differentiation among populations within groups (Fsc = 0.01382) were responsible for 0.84% of the variance. Additionally, the F_ST_ = 0.40362 (*p* < 0.05) indicated low genetic diferentiation among four groups (Table 3).

#### 3.3.3. Historical Demography

A total of fourteen haplotypes forms a network exhibiting a star like topology with the haplotype Hap 1 and Hap 5 at the center (Figure 3). Hap1 was found to be the most widespread, with all 37 populations investigated harboring haplotype Hap 1. A total of 14 populations investigated harbored haplotype Hap 5. Other haplotypes were formed by mutations of these two haplotypes. Hap 1 contains 233 individuals, which might be the ancestor haplotype which evolved into others. In addition, the phylogenetic relationship of all haplotypes based on COI gene was determined (Appendix A). The results showed that all haplotypes from *G. japonicus* formed one cluster and then cluster with *G. swinhonis*. There is no clear genetic lineage formed in populations of *G. japonicus*. Both Tajima’s *D* and Fu’s *Fs* were negative values (Tajima’s *D* = −2.77466, *p* < 0.05, and Fu’s *Fs* = −2.19112, *p* > 0.05, respectively), indicating a distinct departure from a null hypothesis with selective neutrality and population equilibrium. The results of the mismatch analysis of *G. japonicus* did not exclude the sudden expansion model due to non-convergence. The neutrality tests results implied the possibility of population expansion [37,38]. The nonsignificant SSD statistic (SSD = 0.0771, *p* > 0.05) and the *HRag* value (*HRag* = 0.259, *p* > 0.05) under the spatial expansion models failed to reject the spatial expansion model.

## 4. Discussion

In this study, the mitochondrial COI gene was explored for the identification and analysis of the genetic diversity and genetic structure of *G. japonicus* in China. Phylogenetic analyses of *G. japonicus* with 9 other species of geckos were also conducted to clarify the genetic relationship among the genus *Gekko*. Additionally, the genetic diversity and genetic structure of *G. japonicus* in China was evaluated via mitochondrial COI. To the best of our knowledge, this study is the first to examine the mitochondrial genetic diversity of *G. japonicus* in China using the COI gene. Our study contributes a taxonomic dataset and also provides significant insights into the population genetics of an ecologically vital reptile species in China.

The phylogenetic relationship of *G. japonicus* group was well supported, and the *G. swinhonis* were divided into A1, A2, and B. This result was consistent with a previous work using mitochondrial cytochrome *b* gene and 10 microsatellite loci, which revealed two genetic lineages in *G. swinhonis* [39]. Since AHAQ06 clustered with all the *G. japonicus*, we employed genetic distance to demonstrate the reliability of the phylogenetic analysis and the specificity of the AHAQ06. Phylogenetic analyses showed that *G. japonicus* has a close genetic relationship to *G. wenxianensis* in the genus *Gekko* (Figure 2), this result is the same as the one obtained in the previous paper [40]. The current distribution of *G. japonicus* is southern Shaanxi and Gansu provinces in China where it borders *G. wenxianensis*. Hence, their close genetic relationship might be explained by geographic isolation of an independent branch of *G. japonicus* in Wenxian county, which led to speciation, which will give us a new insight into the speciation and dispersal of *G. japonicus*.

The single individual AHAQ06, clustered with all other *G. japonicus* samples, but is genetically distinct, may represent a new species. However, this study is limited by the lack of samples. More samples are needed for further studies of molecular and morphological characteristics.

An overall low level of genetic diversity was detected for *G. japonicus* from China, considering the low *π* and *h*, despite a significant variability of environmental conditions between the geographic regions examined. Genetic diversity in populations is the result of the interplay between mutation, genetic drift, selection, and gene flow [41,42]. Genetic diversity can be lost from populations through various mechanisms. There may be a vital factor responsible for low genetic diversity. Low genetic diversity could be associated with founder and bottleneck events followed by a potential recent population expansion. Demographic bottleneck will increase the action of genetic drift and cause a general loss of genetic diversity, in a magnitude determined by its severity and duration [42,43], as has already been observed in African elephants [44], black-footed ferrets [45], and Arctic foxes [46]. Thus, low genetic diversity showed the possibility of a recent bottleneck for the populations of *G. japonicus* in China, resulting in a smaller original population before an expansion or bottleneck. Time lapse is not sufficient to accumulate sufficient genetic diversity as well as counter the effects of initial demographic bottleneck. In addition, more genetic data, especially the nuclear genome, need to be collected for further studies of the evolutionary history and genetic structure of *G. japonicus*. To better comprehend the genetic diversity of *G. japonicus* at the species level, collecting more samples of *G. japonicus* is therefore highly required.

SAMOVA provided the highest significant F_CT_ (0.39527) when k = 4 was the partition (Appendix A). Subsequently, the AMOVA results demonstrate the distinct genetic structure when populations were grouped. The source of genetic differences emerged primarily from within the populations and rarely from among populations within groups. The absence of variation from among populations within groups may be due to human activities that have promoted gene flow. Previous studies have revealed that contemporary genetic flow is especially high between east coast China and west and central coast Japan, and between west and central coast Japan and Korea. Considering that geckos are frequently invasive species [47,48], it is highly probable that these contemporary mixing processes are a result of anthropogenic transportation [4]. Considering that high commercial trade exists within countries, human-assisted dispersal may have played a vital role in the diversification of *G. japonicus* in China. The genetic differences among groups were at a low level. Among them, Group B and Group D are located at the western and southwestern edges of the Chinese range of *G. japonicus*, respectively. Considering these, we infer that the low level of genetic differentiation between groups is possibly associated with the combination of geographic blockage and gene flow. Despite the human-assisted dispersal factor, the effects of climatic and geographical factors were not sufficient to be countered. Overall, the genetic differentiation of *G. japonicus* was significant, which was likely due to environmental heterogeneous selection. Founder and bottleneck events are further corroborated by the negative values in the neutrality tests (except Fu’s *Fs*). Long-distance dispersal events tend to result in reduced genetic diversity due to sequential founding events and genetic drift [49]. Moreover, the potential population expansion was supported by the “star-like” cluster originating from the haplotypes H1 and H5. Hap 1 might be the ancestor haplotype. Star-like patterns in sequence networks are generally considered to be characteristic of potential recent range expansion [50]. In the present study, the pattern of haplotype distribution is consistent with potential recent range expansion events. Therefore, our study shows that the *G. japonicus* populations have historically experienced founder and bottleneck events causing low genetic diversity. Additionally, phylogenetic relationships between haplotypes show that no clear genetic lineage formed in populations of *G. japonicus*. Seven private haplotypes are found, which may be partially explained by limited migration and gene flow among them.

It is worth noting that the sample sizes of *G. japonicus* are not equal among the groupings in our study and this may have an effect on the diversity measures. Hence, more elaborate studies involving a relatively uniform sample size may uncover the mitochondrial genetic diversity of *G. japonicus* from of China. Admittedly, as our results are based solely on mtDNA sequence data, they should be viewed with some caution, since our conclusions might be biased due to the intrinsic mutation rate, mode of inheritance, and effective population size of mtDNA markers. Thus, further studies integrating ecological, morphological, and molecular (using mtDNA and nuclear markers) data are necessary to understand the evolutionary history of *G. japonicus* from China. Such studies will also give deep insight into the evolution of *G. japonicus* from China. It will further set the base for further genetic studies that would help design effective Ecological Resource Management policies. Also, the genetic approach used in our study can be extended in studying the population genetics of other reptile species in China.

## Figures and Tables

**Figure 1 genes-14-00018-f001:**
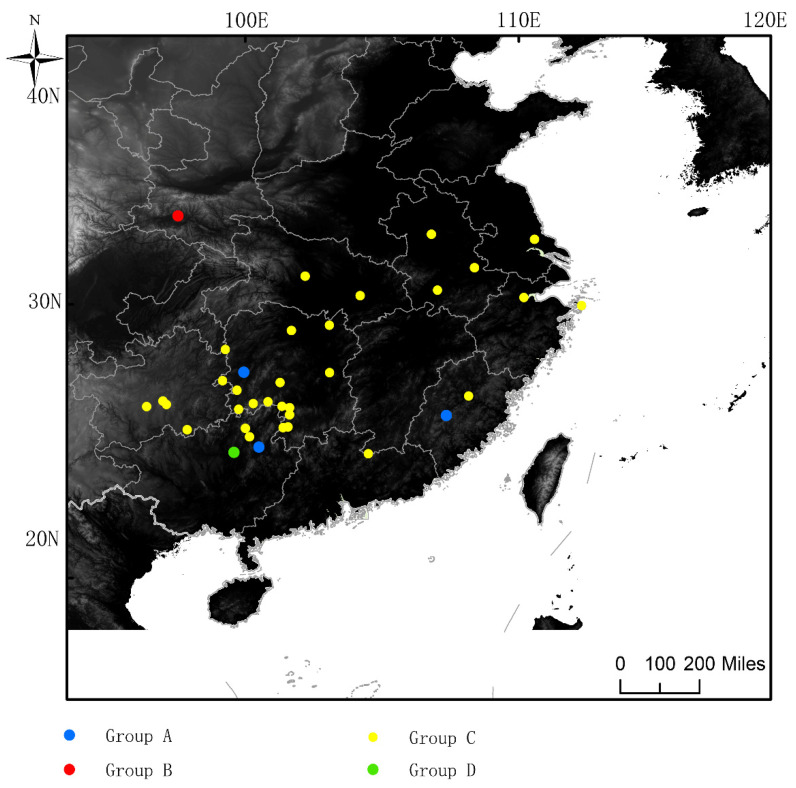
**Sampling sites of *G. japonicus*.** Different colors represent different SAMOVA grouping results The blue circles represent the sampling points of Group A, red circles represent the sampling points of Group B, yellow circles represent the sampling points of Group C, green circles represent the sample sampling points of Group D. Group A: Yongan, Fujian, Yangshuo, Guangxi, Huaihua, Hunan; Group B: Yangxian, Shaanxi. Group C: Anhui (Anqing, Lu’an, Wuhu), Fujian Nanping, Guangzhou Guiding, Guizhou (Huaxi, Longli, Libo), Guangxi (Guilin, Longsheng), Hubei (Jingmen, Wuhan), Hunan (Chengbu, Changde, Daoxian, Huayuan, Huayuan, Shuangpai, Shaoyang, Tong, Xinhuang, Yueyang, Yongzhou, Xinning, Yongzhou Lengshuitan, Zhuzhou, Yongzhou Qingtang Yueyyan Forest Farm), Jiangsu Rugao, Jiangxi Longnan, Zhejiang (Yongzhou Qingtang Yueyyan Forest Farm). Zhoushan, Lishui). Group D: Yongfu, Guangxi.

**Figure 2 genes-14-00018-f002:**
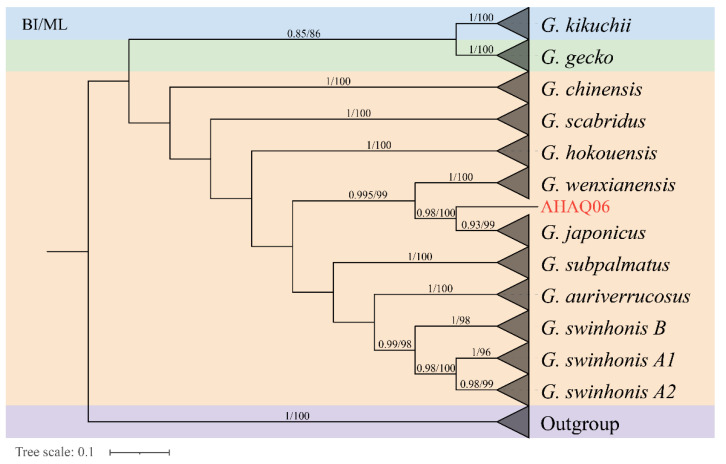
**BI and ML tree constructed based on the COI gene sequence of 11 species and AHA06 of *Gekko* from China.** Members of the *G. japonicus* group species are shown in orange, *Gekko gecko* group species are shown in green, *Gekko monarchus* group species are shown in blue and the purple part is the outgroup. The numbers above branches specify the posterior probability and bootstrap percentages (1000 replicates).

**Figure 3 genes-14-00018-f003:**
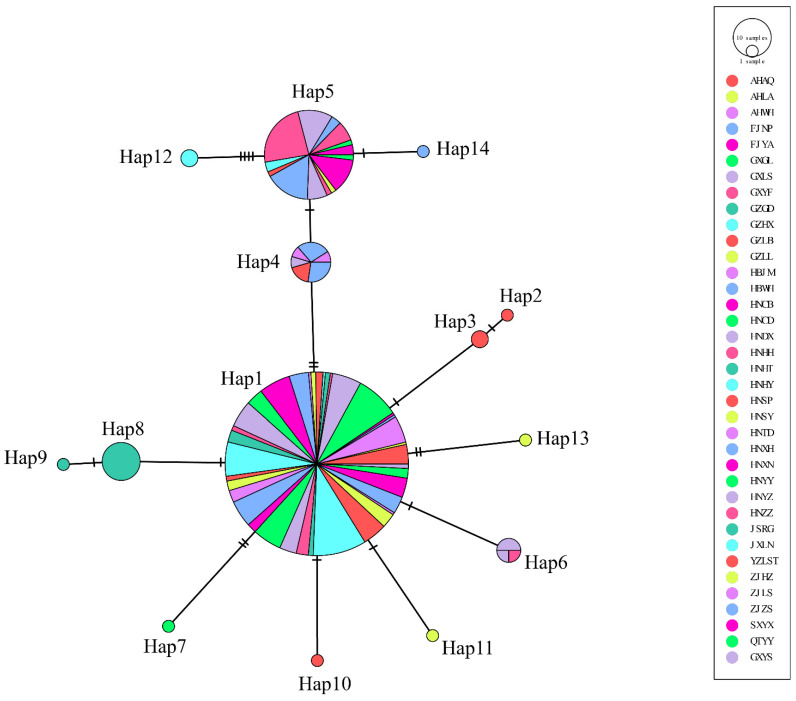
**The relationship diagram of the haplotype network of the *G. japonicus* based on the COI gene.** The circles indicate the haplotypes; circle sizes indicate the probability of haplotypes. The different colors in the circles indicate the distribution in different populations, and the oblique lines indicate mutations between haplotypes.

**Table 1 genes-14-00018-t001:** Analysis of genetic distance within and between species of Chinese *Gekko*.

Species	1	2	3	4	5	6	7	8	9	10	11
1. *G. japonicus*	**0.6 ± 0.2**										
2. AHAQ06	12.0–12.9	**-**									
3. *G.wenxianensis*	21.7–23.5	23.6–24.1	**1.0 ± 0.3**								
4. *G. chinensis*	31.7–33.3	28.3–31.0	25.4–28.7	**3.4 ± 0.6**							
5. *G. swinhonis*	30.0–34.4	27.2–31.6	29.2–32.8	25.1–30.0	**9.1 ± 0.9**						
6. *G. subpalmatus*	31.6–33.0	32.1–33.0	28.4–30.6	29.1–30.2	22.5–25.8	**1.3 ± 0.4**					
7. *G.auriverrucosus*	30.2–31.4	30.5–31.3	31.5–32.7	30.2–33.0	24.0–27.2	24.5–26.2	**0.3 ± 0.2**				
8. *G. hokouensis*	33.5–35.4	29.8–31.3	28.7–30.8	30.6–33.7	28.9–31.4	27.9–30.1	30.5–33.1	**0.9 ± 0.3**			
9. *G. scabridus*	31.4–35.2	33.8–36.2	26.8–30.7	24.1–29.3	23.2–29.5	25.3–29.0	27.3–31.7	29.2–33.0	**4.3 ± 0.6**		
10. *G.gecko*	36.2–38.6	35.1–35.8	35.4–37.3	36.4–38.5	34.2–37.8	42.0–44.9	39.1–40.7	37.0–40.8	35.3–38.5	**2.1 ± 0.4**	
11. *G.kikuchii*	33.0–35.2	35.5	37.3–38.1	28.7–30.4	28.2–33.6	28.1–28.2	36.1–37.3	36.2–37.8	28.5–29.7	32.6–33.4	**-**

Note: The table below the diagonal line indicates the interspecific genetic distance, the bold numbers represent the average genetic distance within each species, and the genetic distance is expressed as a percentage.

**Table 2 genes-14-00018-t002:** Genetic variation parameters of 37 various geographic populations.

Population	Number of Samples	Segregating Site (*S*)	Haplotypes	Haplotype Diversity (*h*)	Nucleotide Diversity (*π*)	Average Number of Difference (*K*)
Anqing, Anhui (AHAQ)	11	2	Hap1, 2, 3	0.473	0.00093	0.618
Lin’an, Anhui (AHLA)	1	0	Hap1	NA	NA	NA
Wuhu, Anhui (AHWH)	12	2	Hap1, 4	0.167	0.00050	0.333
Yong’an, Fujian (FJYA)	3	3	Hap1, 5	NA	NA	NA
Nanping, Fujian (FJNP)	1	0	Hap1	NA	NA	NA
Guilin, Guangxi (GXGL)	19	3	Hap1, 5	0.105	0.00047	0.316
Longsheng, Guangxi (GXLS)	14	1	Hap1, 6	0.264	0.00040	0.264
Yongfu, Guangxi (GXYF)	5	3	Hap1, 5	0.400	0.00180	1.2
Guiding, Guangzhou (GZGD)	2	0	Hap1	NA	NA	NA
Libo, Guizhou (GZLB)	3	0	Hap1	NA	NA	NA
Huaxi, Guizhou (GZHX)	1	0	Hap1	NA	NA	NA
Longli, Guizhou (GZLL)	2	0	Hap1	NA	NA	NA
Wuhan, Hubei (HBWH)	13	3	Hap1, 4, 5	0.590	0.00197	1.308
Hubeijing (HBJM)	1	0	Hap1	NA	NA	NA
Chengbu, Hunan (HNCB)	13	0	Hap1	0	0	0
Changde, Hunan (HNCD)	8	2	Hap1, 7	0.250	0.00075	0.5
Daoxian, Hunan (HNDX)	19	4	Hap1, 5, 6	0.556	0.00237	1.579
Huaihua, Hunan (HNHH)	15	3	Hap1, 5	0.248	0.00112	0.743
Huitong, Hunan (HNHT)	16	2	Hap1, 8, 9	0.542	0.00091	0.583
Huayuan, Hunan (HNHY)	16	3	Hap1, 5	0.233	0.00105	0.7
Shuangpai, Hunan (HNSP)	4	4	Hap1, 5, 10	NA	NA	NA
Shaoyang, Hunan (HNSY)	5	1	Hap1, 11	0.400	0.00060	0.4
Tongdao, Hunan (HNTD)	6	2	Hap1, 4	0.333	0.00100	0.667
Xinkou, Hunan (HNXH)	20	3	Hap1, 5	0.521	0.00235	1.563
Xinning, Hunan (HNXN)	4	0	Hap1	NA	NA	NA
Yueyang, Hunan (HNYY)	12	0	Hap1	0	0	0
Yongzhou, Hunan (HNYZ)	12	3	Hap1, 4, 5	0.591	0.00232	1.545
Zhuzhou, Hunan (HNZZ)	7	4	Hap1, 5, 6	0.524	0.00172	1.143
Rugao, Jiangsu (JSRG)	2	0	Hap1	NA	NA	NA
Longnan, Jiangxi (JXLN)	24	6	Hap1, 12	0.159	0.00144	0.957
Yueyan Forestry, Qingtang, Yongzhou, Hunan (QTYY)	5	3	Hap1, 5	0.400	0.00188	1.200
Yangxian, Shaanxi (SXYX)	15	3	Hap1, 5	0.533	0.00241	1.600
Lingshutan, Yongzhou, Hunan (YZLST)	12	2	Hap1, 4	0.303	0.00091	0.606
Hangzhou, Zhejiang (ZJHZ)	8	5	Hap1, 5, 13	0.464	0.00188	1.25
Zhoushan, Zhejiang (ZJZS)	11	4	Hap1, 4, 14	0.564	0.00208	1.382
Lishui, Zhejiang (ZJLS)	1	0	Hap1	NA	NA	NA
Yangshuo, Guangxi (GXYS)	1	0	Hap1	NA	NA	NA
Overall	324	16	14	0.453	0.00176	1.171

**Table 3 genes-14-00018-t003:** AMOVA analysis of the *G. japonicus* population.

Source of Variations	Sum of Squares	Variance Components	Percentage of Variation	Fixation Indices	Significance Tests
Among groups	47.762	0.57420	39.53	F_CT_ = 0.39527	0.00000
Among populations within groups	32.026	0.01214	0.84	F_SC_ = 0.01382	0.02639
Within populations	252.108	0.86635	59.64		
Total	333.896	1.45269		F_ST_ = 0.40362	0.03030

## Data Availability

All raw sequencing reads have been deposited in the NCBI Sequence Read Archive under project SUB12126169. This article contains supplementary figures and tables, which are available to authorized users.

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
