# Peer review of "Phylogenetic Analysis and Genetic Structure of Schlegel’s Japanese Gecko (Gekko japonicus) from China Based on Mitochondrial DNA Sequences"

_genes, 2022, doi:10.3390/genes14010018_

Round 1

Reviewer 1 Report

The objective of this study seemed ambiguous and inconsistent, and there seemed some meaningless analyses and results. On the other hand, some of the results were potentially valuable and interesting in the context of taxonomy and phylogeography: 1) the (preliminary) discovery of a putative cryptic species from Anqing represented by the specimen AHAQ06; 2) the population genetic characteristics of Gekko japonicus; and 3) morphological difference between clades A and B of G. swinhonis. I think this paper can be restructured to focus on 1) and 2), or 1) 2) and 3).
Besides, the writing of the manuscript was entirely not very good. The materials and methods and results including figures and tables are too complicated and difficult to understand. See the comments in the extra file to the author for further details. In addition, its English needs careful verification (there are some basic grammatical errors, for example lines 378--379 and 404) and following linguistic correction by a native speaker (maybe using a commercial service).

Reviewer 2 Report

The following is a review of the manuscript “The mitochondrial COI gene analysis indicates low genetic diversity and clear genetic structure in Gekko japonicus” for the journal Gene.

In this paper, the authors use sequence data of the COI mtDNA gene from a large number of Gekko japonicus samples throughout southwestern China. They compare morphology for specimens of this species with closely related species based on their mtDNA analyses. They also conduct demographic analyses and population structure, within G. japonicus. They identify a potentially new lineage, and confirm previous studies that G. swinhonis may represent two species. However, they do not make any formal taxonomic recommendations. They infer they ancestral area of G. japonicus to be Anqing City, Anhui Province, China. However, the type locality for G. japonicus is Japan, and they currently lack samples from Japan. The paper has a number of other errors and grammatical errors and some conclusions that are a bit beyond the results they present. I think it could be considered for publication with significant revisions. The tree figures area barely legible and poorly presented.

First, they indicate they are conducted “DNA barcode analyses” (e.g., bottom page two, lines 74–80). They cite some papers on agamid lizards that use mtDNA to conduct phylogenetic analyses; not DNA barcoding, this term gets used loosely. They then cite Hebert et al., formal DNA barcoding. However, they only present analyses and results for the ABGD analyses, an alternative to the standard barcoding analyses (e.g., BINs).

They describe in detail, the morphology of their identified putatively new species, yet they cite the more rigorous threshold criteria (lines 371–372) for genetic varition as not being reached. It is unclear where this threshold comes from and why, here in the Discussion, and why they are adhering to it. They should consider describing this as a new species, 12% seq divergence is WAY BEYOND the accepted divergence between vertebrate species with DNA barcode data (typically between 2–3% is species-level).

They should present all of the molecular results together and the morphology together, not with the morphology in between molecular results (as currently arranged). Also, it would make more sense to provide the within G. japonicus variation first, saying how many haplotypes, etc., then present the unique haplotypes in the phylogeny of the genus. Also, Gekko was recently broadened to contain other genera (e.g., Ptychozoon). The authors here should use a more inclusive group name “G. japonicus group?”

Figure 2 and 3 are poorly constructed and not legible. They need to make the font larger on the tips of the trees, to determine which samples are represented. One major problem is that it appears they included all of the identical haplotypes of G. japonicus. They have only 15 haplotypes of G japonicus. They only need one representative of each haplotype. In fact, RAxML and IQ-Tree remove redundant haplotypes in the analyses and provide a truncated dataset. They should use for their figure. It is clear from their sample, the map in Fig. 1, and the tables, that they have many individuals. They can simply use H1–15 in their tree that will match their Table 2 (or include individuals for private haplotypes).

It is unclear how they determine the center of origin for G. japonicus in Anhui Province, China, when they lack material from the type locality, or any samples from Japan, or Korea. They discuss low genetic diversity with Chinese G. japonicus, as a bottleneck, founder effect, this contradicts China as a place of origin.

How were the genetic distances calculated? Doesn’t say in the Results (line 201) or Table 1.

I think the last three paragraphs (highlighted green) in the Discussion are speculation beyond the results of the present study. I think this should be deleted, unless they can demonstrate more clearly how they suspect that Anhui Province, China is the center of origin for G. japonicus.

Many of the references use an initials-only abbreviation for the journals, after the list of authors. I have never seen this before. Either way, it is inconsistent with other articles cited.

Detailed comments:

Abstract:

Line 9 delete “which originates in China”,

Line 18 replace “were supported” with “is provided”.

Delete lines 20–22 (from “Relatively clear… to … origin for G. japonicus”)

Intro:

They make statements that require citations;

e.g., line 9 “which originates in China”,

lines 35–38. “Prediction of present and future distribution studies which were based on… suggest that…” No ref. at end, and how can “future distribution studies” “which were based on…” be assessed?

And line 365 “Subsequent evolutionary analyses also support this opinion” where?

Line 58 replace “retrievals” with “identifications”

Line 65 delete “Cryptic species are identical in morphological grounds” (already stated this above).

Line 87 replace “reconstructed” with “conducted”.

Line 88 delete “possibly”

Line 89 delete “the genetic structure of G. japonicus in China. Further,

Line 90 replace “were evaluated” with “in China”

Materials methods

Lines 159–160 move to Results section.

Results

Lines 186–187 should read: “…showed that all species formed monophyletic groups each with high bootstrap…”

Line 189 replace “Distinguishingly” with “The twelfth”

Line 191 add “samples” to read “…that the G. swinhonis samples were separated…”

Lines 192–193 move to Discussion.

Line 196 delete “in the genus Gekko”

Line 197 should read “… and the two formed a sister group with G. swinhonis.” (full stop.

Line 200 replace “studied” with “supported”

Line 219 replace “According to” with “Based on the” and replace “the experiment” with “this study”.

Line 226 replace “Difference result” with The different results”.

Line 228 delete “, further proof of its specificity (this is first mention of it being a species).

Line 230 italicize G. japonicus

Line 244 The variation is not explained well here, or in the figure S1.

Lines 246–247 should read “… at the base and 17-27 precloacal and femoral pores were obvious… characteristics … that distinguish it from G. japonicus.”

Line 249 delete “with”

Line 252 replace “belong” with “belonging”

Line 253 replace “, with” with “had”

Line 258, they say “…clade B is described in conjunction with our lab in this study” this is not true, it is not formally described in this current study. Remove.

Line 263 replace “webbed” with “webbing”

Line 265 replace “Besides” with “Additionally,” and delete “the” in front of “G. swinhonis”

Line 278 should read “was compared morphologically with specimens of G. japonicus.”

Line 293 “G. japonicus” needs to stand out, either not italics or underlined.

Line 340 replaced “used” with “depicted”

Discussion

Line 350 replace “exploited” with “explored”

Line 352 should read “DNA barcoding provided an accurate and rapid species identification system, and was combined with…”

Line 353 replace “Meanwhile” with “Additionally,”

Lines 355–356 delete sentence “Posteriorly, the genetic…COI genes.”

Lines 362–363 delete “, in addition to a few morphological variation phenomena,”

Line 364 delete “, which”

Line 365 delete “Subsequent evolutionary analyses also support this opinion” (or provide reference).

Line 365 replace “Larger” with “Large”

Lines 366–367 Why describe it morphologically, in such detail, if not recognizing it as a species?

Line 378 replace “to southern” with “is southern”

Lines 382–384 delete.

Reviewer 3 Report

The titled "The mitochondrial COI gene analysis indicates low genetic di-2 versity and clear genetic structure in Gekko japonicus" manuscript presents population genetic patterns of these species in China. Although it has been studied in its wide geographic distribution (Kim et al., 2020), it was not came across its population genetic study in specific to China. Therefore, I think this study will be published by the journal. However, it needs to some revisons. My all comments and corrections added on the pdf file.

Kim, J.-S.; Park, J.; Fong, J.J.; Zhang, Y.-P.; Li, S.-R.; Ota, H.; Min, S.-H.; Min, M.-S.; Park, D.J.M.D.P.A. Genetic diversity and 510 inferred dispersal history of the Schlegel’s Japanese Gecko (Gekko japonicus) in Northeast Asia based on population genetic 511 analyses and paleo-species distribution modelling. 2020, 31, 120-130.

Reviewer 4 Report

This manuscript aim to explore genetic diversity and population structure of G. japonicus in China using COI sequences. However, when I went through the manuscript in detail, the manuscript is more likely be divided into two main parts: (1) phylogenetic relationship of Chinese Gecko based on combined morphological and COI analyses plus (2) genetic diversity and phylogeography of G. japonicus. I think it is possible that authors will have two main relevant objectives in one manuscript but authors need to organized the story well. If not, authors will make reader confused about the story.

I have several comments on your manuscript, please see the attached and some major comments are listed down here. 

Abstract:

It is well written, but no results about phylogenetic analyses and morphological observation. Please add some of those results into this part. 

Introduction

L51 - 66, I think this paragraph is unnecessary to show here. It does help much to get more detail that related to your story. Please reconsider.

L83-92, I think authors need to rewrite and reorganize this paragraph. It is important paragraph to let the reader know clearly about research aims.

Material and Methods 

2.2.1 Phylogenetic analyses 

- Did author use the AIC for both ML and BI methods? Why don’t author apply BIC for BI analyses?

- Why did author use only 100 replications for ML ? Is it reliable? Normally 1000 replications, isn’t it?

2.2.2 Species delimitation 

- I suggest that authors should perform not only ABGD, but also include other method, for example GMYC and bPTP.  

- Why did authors set minimum gap of 1.5 P? I think this value can be specified based on p-value. Please reconsider about setting the minimum value.

2.2.3 Morphological observation

I think this subtopic should be raised as another topic, not under 2.2 DNA extraction and sequencing. 

I am just wondering why didn’t author show even few morphology of G. japonica, even authors carefully observed their morphology. If authors want to report about morphological characteristic, figure of detailed morphology is really needed. Please add all figures describing some key morphological features of those species. 

I also recommend that authors ,if still need to deal with morphological issues, must draw one another table to compare some key characters among G. japonica and related species. 

2.2.4 Population genetic analysis

I just wondering how did authors perform AMOVA for population with one to five samples compared with another populations with more than 1o samples. Can we do that ???

Results

3.1 Phylogenetic analyses

- I think author should show only one tree with both ML and BI bootstrap on each the clade. Also authors need to edit your tree. Frankly its present form is unacceptable, and it is really difficult to see in detail. It would be nice if author can collapse a clade and make cartoon tree.

3.2 ABGD

- I think you need to combined ABGD results with phylogenetic tree, it will make reader understand more clearly about the results. Frankly, I don’t understand clearly by your writing. Please refer to Laopichienpong et al. (2016) (https://www.sciencedirect.com/science/article/pii/S0378111916307338#f0020)

3.3 Morphological comparison

Please see above comments (2.2.3). 

Frankly, I am really confused about what is the main objective of this study. The tittle is about genetic diversity and population structure of G. japonicus, but authors also worked on morphological observation, phylogenetic relationship and species delimitation. I think that title is not fully related to your story. I suggest that author need to reconsider and please more focus for what you want to present. It is likely that authors want to explain all Gecko species in China, not only G. japonicus.

3.4 Genetic Diversity

- Table 2 should be merged with Table 3. 

- Figure 4 can be removed. No need at all.

- Mismatch distribution showed steep peak ??? I think it normally showed unimodal or multimodal. Please carefully interpret the results.

- “Hap 1 and Hap 6 were 339 used as the central radiation distribution” Why Hap 1 and 6, not Hap 2 and Hap 4 (Figure 5). What is the definition of central radiation distribution ?????

- Figure 5, what did Group A, B, C D represent ????

Discussion 

Good luck

Round 2

Reviewer 2 Report

The following is my second review of the manuscript “Phylogenetic analysis and clear genetic structure of the Schlegel’s Japanese Gecko (Gekko japonicus) from China based on mitochondrial DNA sequences” for the journal Gene.

In this paper, the authors use sequence data of the COI mtDNA gene from a large number of Gekko japonicus samples throughout southwestern China. They compare morphology for specimens of this species with closely related species based on their mtDNA analyses. They also conduct demographic analyses and population structure, within G. japonicus. They identify a potentially new lineage (AHAQ06), and confirm previous studies that G. swinhonis may represent two species. However, they do not make any formal taxonomic recommendations.

Given that they are not describing the new species identified by AHAQ06, they go into a very detailed description of its morphology. I understand that they do not want to describe it based on this single individual. They make statements that by providing a detailed morphological description, it will make describing this species in the future easier. However, I do not think this is true. They will still have to provide a detailed description of the morphology in a formal species description, and they will presumably have additional material, which will change the morphological description. Therefore, I think they should reduce this part of the manuscript, they should not describe it in such detail (as a species description), but instead just simply describe how it differs from G. japonicus.

Otherwise, they have made several improvements to the manuscript, but there are still a number of grammatical errors. I think it could be considered for publication with minor revisions. The tree figures are much better.

Specific comments:

Line 2: Title remove “the” in front of Schelgel’s Japanese Gecko.

Line 18, add “and” in front of expressed a relatively…

Line 28: add “be”  to read “to be distributed…”

Line 30, those papers cited [2,3] do not analyze mitochondrial DNA; therefore, remove “and mitochondrial genome,” or cite additional references.

Line 52 remove “taxonomy”

Line 56 change “morphological taxonomy” to “morphology” (taxonomy doesn’t provide evidence for classification, it is a result of classification; morphology can provide strong evidence for classification).

Line 61 change to read “…especially mitochondrial DNA, has been shown…”

Line 64 add “studies” to the end of the sentence “…phylogenetic studies.”

Line 70-71: change to read “…of G. japonicus, 325 specimens were collected from 37 sampling…”

Line 72, add “these” to read “…from these 325 specimens in China”

Line 72, use lower case for “phylogenetic analyses”

Line 78, add “the” mitochondrial gene COI.

Materials and Methods

Line 87 add “that” was used…

Figure 1. Caption. Replace “mentioned” with “used” and replace “paper” with “study”.

Lines 91-93 remove “sample” (3x) because “sample sampling points” is redundant.

Line 118 remove “the” GenBank

Line 119 remove “the” previously…

Line 122 remove period in parentheses, and remove “genes” at end of sentence.

Line 127 Either replace the period with a comma, after “(Table S2)” or remove “in which” and start new sentence with “Hemidactylus dracaenacolus and H. granti were used as outgroups…” (replace “utilized” with “used”)

Line 168 add “by” to “employed by the bootstrap method.”

Line 169, what is M. vulgaris? I think this was in error.

Results

Line 176 replace “was depicted” with “of G. japonicus were analyzed”

Line 181 fix “analysi”

Line 185, unclear what G. monarchus group is? What species in their tree are in this group?

Line 186, remove “lineage” and replace with “separated into three lineages, A1, A2, and B.”

Line 189, lower case “A total of (follows a comma).

Figure 2 caption, line 206 switch order of posterior probs and bootstrap to match order in figure.

Line 212 italics for G. japonicus

Line 213, delete “the” and change “has” to have, to read “…characteristics of G. japonicus have been…”

Line 218 add “and” to “…, and have a higher…”

Line 222-3, remove “in” and replace “characters” with “morphologically” to read “…were examined morphologically.”

Line 236, replace “are” with “is” and add “population-level” to read “Hence, AHAQ06 is excluded from the population-level data set.”

Line 251 add “s” to the end of population and remove double period.

Line 252, why would they compare geckos in China to a Puerto Rican snake?

Line 257 replace “sequences” with “data”.

Line 259, reword to “Four groups were separated as follows:”

Line 266, reword to “The total variation was…”

Discussion

Line 303, remove “the” in front of G. japonicus

Line 304, delete “Meanwhile,” and start with “Phylogenetic analyses … were “also” conducted….

Line 306-7 remove “the” in front of G. japonicus” and remove “via mitochondrial COI genes” or remove “genes”.

Line 312 remove “branches”

Lines 314-316, I think they should move this to the beginning of the next paragraph, as a transition to the next paragraph, and change it to something like: “A single individual (AHAQ06) clustered with all other G. japonicus samples, but is morphologically and genetically distinct, we suspect it may represent a new species.”

Lines 321-22: rewrite this sentence to something like “Hence, their close genetic relationship might be explained by geographic isolation of an independent branch of G. japonicus in Wenxian county, which led to speciation.”

Line 323, remove “Worth mentioning is” and start this paragraph with the sentence mentioned above (“A single individual…” Then change this sentence to something like, “This specimen was considered a separate species and removed from subsequent phylogenetic and genetic distance analyses of G. japonicus.”

Lines 325-327 remove the sentence “Then, we conducted a detailed morphological description…”

Line 329, rewrite to something like, “… is limited by the lack of samples. More samples are needed for further…”

Line 330 remove this sentence, because it is basically stated above that is was removed from the subsequent analyses.

Line 335, remove “recombination” (doesn’t happen in mtDNA).

Line 341, “As has already…” not a complete sentence, either replace period at the end of the previous sentence with a comma and use lower case “as”, or add replace “As” with “This” has already been observed…” though, I’m not sure showing this has happened in elephants, foxes, and ferrets is really warrented.

Line 346, remove “data” after genome.

Line 380 replace “are” with “were” found, …

Line 385 remove “further,”

Reviewer 4 Report

I think that this manuscript still need to revise more carfully. I found lot of flaw. Please see comments below. 

Abstract

Authors mentioned that “G. japonicus populations were divided into four  groups and exhibited moderate levels of genetic differentiation”, and I think the results from Table 4 indicate Fct=0.39, suggesting low genetic differentiation. I don’t think it is moderate level. Also I did not see any clear pattern from your data (Fig. 1).  

Group A, C and D are grouped together, and Group B is likely to be isolated from “other group”. How did author obtain this results from SAMOVA? I think Fig 1 are really congruent with AMOVA analyses that indicate low genetic differentiation among group and no clear pattern. 

I also would like to ask about what evidences indicate of population expansion of G. japonicus in China as authors mentioned in abstract. However, I quite believe the recent expansion of this species based on low genetic diversity, low genetic differentiation and unclear population structure.

Please carefully reinterpret the results.

Introduction

I think that the second paragraph is too long and it is a bit out of scope. I think author do not need to emphasize much detail in taxonomic status and morphology of this species, 

M&M

Just wondering how to make phylogenetic relationship of haplotype using PopART ???

Results & Discussion

I recommend to remove AHAQ06 from this study. I think AHAQ06 could potentially be new species (12% genetic difference is quite huge….I think), so author should sampling more and work carefully in its morphology and genetic analyses to clarify the taxonomic status of this samples.

I don’t understand clearly about steep peak of mismatch distribution results, as normally this analyst will show unimodal, indicative of demographic expansion, and multimodal, indicative of stable population. I already mentioned about this at my last comment, but no response. 

I also don’t understand what is the purpose of Figure 4. Authors mention about Figure 4 and SAMOVA….what did two these things relate to each other ??? 

Are authors serious to include figure 4 (phylogenetic relationship of COI haplotypes) in this manuscript. I do not feel necessary for showing this figure. I think author should combine fig 2 and 4 as one and put it in supplementary document.

I also suggest to remove morphological observation from this paper. 

How reader recognize of the difference of color in figure 3? Impossible, isn’t it?

I don’t support the idea of 4 group within this species as I mentioned above, please reconsider about the results. 

I will suggest author to focus on genetic diversity and population structure on G. japonicus population in China. I don’t think you need to include AHAQ06 in your analysis. 

Also please check the grammar and misspelling….I found it throughout the manuscript. 
